# Comparative Research on Metabolites of Different Species of *Epichloë* Endophytes and Their Host *Achnatherum sibiricum*

**DOI:** 10.3390/jof8060619

**Published:** 2022-06-10

**Authors:** Yongkang Deng, Yuan Gao, Chenxi Li, Junzhen Zhang, Xiaowen Fan, Nianxi Zhao, Yubao Gao, Anzhi Ren

**Affiliations:** College of Life Sciences, Nankai University, Tianjin 300071, China; 2120201009@mail.nankai.edu.cn (Y.D.); 2120140951@mail.nankai.edu.cn (Y.G.); 2120201012@mail.nankai.edu.cn (C.L.); 1120190472@mail.nankai.edu.cn (J.Z.); 2120190969@mail.nankai.edu.cn (X.F.); zhaonianxi@nankai.edu.cn (N.Z.); ybgao@nankai.edu.cn (Y.G.)

**Keywords:** *Epichloë* endophyte, *Achnatherum sibiricum*, GC–MS, metabolite, correlation

## Abstract

*Achnatherum sibiricum* can be infected by two species of fungal endophytes, *Epichloë* *gansuensis* (Eg) and *Epichloë* *sibirica* (Es). In this study, the metabolites of Eg, Es, and their infected plants were studied by GC–MS analysis. The results showed that the metabolic profiles of Eg and Es were similar in general, and only six differential metabolites were detected. The direct effect of endophyte infection on the metabolites in *A. sibiricum* was that endophyte-infected plants could produce mannitol, which was not present in uninfected plants. *Epichloë* infection indirectly caused an increase in the soluble sugars in *A. sibiricum* related to growth and metabolites related to the defense against pathogens and herbivores, such as α-tocopherol, α-linolenic acid and aromatic amino acids. *Epichloë* infection could regulate galactose metabolism, starch and sucrose metabolism, tyrosine metabolism and phenylalanine metabolism of host grass. In addition, there was a significant positive correlation in the metabolite contents between the endophyte and the host.

## 1. Introduction

Fungal endophytes colonize the intercellular space of plant aboveground tissue and complete all or part of the life cycle in host plants [1]. Fungal endophytes are particularly common in Gramineae [2], and generally have host specificity [3,4]. Fungal endophytes in grasses have been reported to belong to many genera, but *Epichloë* endophytes of Clavicipitaceae (Ascomycetes) have been extensively studied, and approximately 30% of cold-season grasses could be infected with *Epichloë* endophytes [5]. A number of studies have shown that *Epichloë* endophytes and host grasses form a close symbiotic relationship of mutual benefit. Host plants provide nutrients and places for the survival and reproduction of endophytes [6], and endophytes can not only promote the growth and development of host grasses [7,8], but also improve the tolerance of host plants to abiotic stresses, such as drought, high temperature, saline-alkali and heavy metals [9,10,11,12,13], as well as resistance to biotic stresses, such as pathogens and herbivores [14,15,16,17], and enhance the intraspecific and interspecific competitiveness of host plants [18,19].

The effects of fungal endophytes on the growth and stress resistance of host plants may be related not only to the active metabolites produced by themselves but also to the change in metabolism in host plants caused by endophyte infection. In previous studies of active metabolites produced by *Epichloë* endophytes, only alkaloids have been proven to protect hosts from herbivory. Four types of alkaloids have been reported: ergot alkaloids, indole-diterpene alkaloids (such as lolitrem B), lolines and peramine, all of which have anti-insect activity, and the first two types can cause poisoning in livestock [20,21,22,23,24,25]. The kinds of alkaloids produced in different grass–endophyte symbionts vary with the species of fungal endophytes; for example, *Epichloë festucae* var. *lolii* in perennial ryegrass (*Lolium perenne*) can produce peramine, ergovaline and lolitrem B [14,26], *Epichloë funkii* in *Achnatherum robustum* produces ergot alkaloids and indole-diterpene alkaloids [24], and *Epichloë inebrians* in *Achnatherum inebrians* only produces ergot alkaloids [27], while some *Epichloë* endophytes do not produce alkaloids [21]. In addition, some studies have found that *Epichloë* endophytes can produce some antifungal active metabolites when cultured separately in vitro, including gamahonolides A and B, gamahorin, chokols A–G, indole derivatives and epichlicin [28,29,30,31,32]. However, it has not been reported whether these active metabolites can also be produced in grass–endophyte symbionts.

On the other hand, previous studies on the effects of endophyte infection on the metabolites of host plants were mainly based on the changes in one or several certain plant metabolites under stress conditions. For example, *Epichloë* infection can increase soluble sugars in *Lolium perenne* under drought stress [33], increase proline in *Hordeum brevisubulatum* under salt stress [34], increase salicylic acid, 3-hydroxypropionic acid and glucose-6-phosphate in *Achnatherum inebrians* under low-nitrogen stress [35], and increase the content of phenolic compounds in *Lolium perenne,* when infected with pathogenic fungi [36]. While beneficial fungal endophytes promote growth or improve the stress tolerance of host plants, the effect is not caused by a change in a certain compound but the comprehensive result of broad regulation of metabolism [37]. Therefore, it is essential to analyze the change in plant metabolic profiles through metabolomics methods to systematically reveal how beneficial fungal endophytes affect the metabolism of host plants.

*Achnatherum sibiricum* is a perennial, sparse bunchgrass that is a common companion species in the Inner Mongolia steppe; in the natural population, its *Epichloë* infection rate is commonly close to 100% [38]. The other two species of *Achnatherum* infected with *Epichloë*, *A. inebrians* is distributed in the natural grassland of China and *A. robustum* is distributed in the American continent, and both are associated with livestock poisoning [39,40]. In contrast, *A. sibiricum* infected with *Epichloë* produces only a small amount of peramine after clipping, which is not enough to be toxic to herbivores [41]. Our previous studies found that *Epichloë* infection can promote growth and enhance resistance to herbivorous insects and pathogens of *A. sibiricum* [41,42,43]. We chose to analyze the metabolites in *A. sibiricum* and *Epichloë* because this grass in Inner Mongolia naturally hosts either *E. gansuensis* (Eg) or *E. sibirica* (Es) [44], so it is an ideal material to analyze the relationship of metabolites between fungal endophytes and host grasses. In this study, the metabolites of two *Epichloë* species and the host *A. sibiricum* were analyzed by gas chromatography–mass spectrometry (GC–MS), and three scientific questions were addressed: (1) Does *Epichloë* infection change the composition of metabolites in *A. sibiricum*? (2) Which metabolic pathways of *A. sibiricum* are affected by *Epichloë* infection? (3) Is there a correlation between *Epichloë* endophytes and host *A. sibiricum* metabolites?

## 2. Materials and Methods

### 2.1. Seed Source

The seeds of *A. sibiricum* were collected from the National Hulunbuir Grassland Ecosystem Observation and Research Station (119.67° E, 49.10° N) at the Chinese Academy of Agricultural Sciences. The fungal endophyte infection status of the seeds was determined by the aniline blue-lactic acid staining method, and the infection rate was 100%.

### 2.2. Isolation of Epichloë Endophytes

Five seeds from the same spike were inoculated on potato dextrose agar (PDA) medium and cultured at 25 °C in the dark after surface sterilizing with 50% H_2_SO_4_ and 3% NaClO solution. When the mycelium grew from the surface or edge of the seeds, the mycelium was picked and purified. The process was repeated 3 times to obtain single-cell colonies. The species of *Epichloë* endophytes were identified by means of morphology and phylogeny, which have been reported by Zhang et al. [44]. Then, the mycelium was sampled.

### 2.3. Plant Materials

Seeds from the same spike were infected with the same *Epichloë* species due to vertical transmission [44]. We used seeds from two plants, one harboring Eg and the other harboring Es. To obtain endophyte-free (E−) seeds, Eg-infected and Es-infected seeds were treated with high temperature at 60 °C for 30 days, and the endophyte could be effectively killed. This disinfection method has no significant effect on the seed germination rate of *A. sibiricum* [45]. The E−, Eg-infected and Es-infected seeds were sown in individual plastic pots (20 cm in diameter and depth, respectively) filled with vermiculite, and each treatment had 5 biological replicates for analysis. The pots were placed in the greenhouse at Nankai University, Tianjin, China. Four weeks after germination, 10 similar seedlings were retained in each pot. The plants were supplied with water and Hoagland nutrient solution as needed. Twelve weeks later, leaves with similar growth states were sampled, frozen in liquid nitrogen, and then stored at −80 °C. Endophyte status was checked at the beginning and end of the experiment following staining with lactophenol containing aniline blue [46].

### 2.4. Extraction and Derivatization of Metabolites

The method of extraction and derivatization of metabolites was slightly modified from Lisec et al. [47]. The fungal mycelium or plant leaves were fully ground in liquid nitrogen, and 20.0 mg of sample was transferred into a 2 mL tube. Then, 1400 μL precooled methanol and 100 μL ribitol (2 mg/mL in distilled water) as the internal standard were added to the samples, and the samples were vortexed for 30 s to be fully mixed. The extraction was carried out in a water bath at 70 °C for 15 min. After centrifugation at 11,000 rpm for 10 min, the supernatant was transferred into new tubes, 750 μL chloroform and 1500 μL distilled water were added in turn and then vortexed and centrifuged at 11,000 rpm for 10 min to be stratified. Then, 250 μL supernatant was transferred into new tubes, and the solvent was completely volatilized with a vacuum centrifugal concentrator. Next, 80 μL methoxyamine hydrochloride (20 mg/mL in pyridine) was added, and oximation was carried out at 37 °C for 90 min on a shaker (200 rpm). Finally, the samples were derivatized by 80 μL MSTFA reagent at 37 °C for 30 min on a shaker (200 rpm).

### 2.5. Metabolite Analysis by GC–MS

The samples were analyzed by GC–MS method. As such, 1 μL sample was injected into gas chromatograph (7890A, Agilent, Palo Alto, CA, USA), which was connected to GC capillary column (HP-5MS, 30 m × 250 μm × 0.25 μm, Agilent, Palo Alto, CA, USA). The carrier gas was helium (99.99%), and the column flow was 1 mL/min in nonshunt mode. The initial temperature was set at 70 °C and increased to 310 °C with a rate of 5 °C/min. The temperature was kept at 310 °C for 5 min. The detector temperature was set at 280 °C. The EI ion source temperature was 230 °C, and the electron energy was 70 eV. The acquisition of mass spectrometry data was in full scanning mode with m/z range of 40–510 amu. The peaks of compounds were identified by the NIST11 library, and the peaks with matching degrees ≥80% were screened for analysis. The relative content of each metabolite was expressed by the ratio of the peak area of the metabolite to the peak area of the internal standard [48].

### 2.6. Statistical Analysis

To analyze the difference of effect in *Epichloë* species on the metabolic profiles of endophytes and host plants, the data were examined by principal component analysis (PCA) on Wekemo Bioincloud (www.bioincloud.tech (accessed on 26 February 2022)). Differential metabolites between groups were screened by orthogonal partial least squares discriminant analysis (OPLS-DA) with SIMCA 14.1 software, the date was normalized by unit variance (UV) scaling, and compounds with a variable importance in the projection (VIP) value greater than 1 and a *p* value (*t*-test) less than 0.05 were considered as differential metabolites. Pathway analysis was performed on MetaboAnalyst 5.0 (www.metaboanalyst.ca (accessed on 7 April 2022)) based on the KEGG database of *Oryza sativa* Japonica to analyze the significantly changed metabolic pathways of *A. sibiricum*. In addition, correlation analysis of metabolites of *Epichloë* endophyte and *A. sibiricum* was performed with Microsoft Excel 2013, the fold change (FC) was calculated from the ratio of the metabolite content of Es to Eg (endophyte) and Es+ to Eg+ (plant).

## 3. Results

### 3.1. Metabolites of Epichloë Endophytes

A total of 35 metabolites detected by GC–MS in the mycelium of the two *Epichloë* species were identified (Appendix A), including 11 amino acids, 5 sugars, 7 sugar alcohols, 4 organic acids and 8 fatty acids, among which mannitol had the highest content. The score plot of PCA showed that PC1 and PC2 explained 20.66% and 17.13% of the total variance, respectively, and the metabolic profiles of the two *Epichloë* species were similar in general (Figure 1). Six differential metabolites between Eg and Es were screened by OPLS-DA (VIP > 1, *p* < 0.05). The contents of tyrosine, trehalose, oleic acid and α-linolenic acid in Es were significantly higher than those in Eg, while the contents of proline and glutamic acid in Eg were significantly lower than those in Eg (Figure 2).

### 3.2. Metabolites of A. sibiricum

A total of 38 metabolites detected by GC–MS in leaves of *A. sibiricum* were identified (Appendix A), including 11 amino acids, 9 sugars, 3 sugar alcohols, 8 organic acids, 5 fatty acids, 2 sterols and α-tocopherol, in which mannitol was only detected in endophyte-infected *A. sibiricum*. As shown in the score plot of PCA (Figure 3), PC1 and PC2 explained 35.48% and 10.91% of the total variance, respectively, and the metabolic profiles of *A. sibiricum* infected with Eg and Es were obviously distinguished from those of E plants. Endophyte infection significantly increased 11 metabolites in the hosts, including sucrose, glucose, fructose, allose, maltose, phenylalanine, tyrosine, mannitol, threonic acid, α-linolenic acid and α-tocopherol, while it significantly decreased four metabolites, including valine, ornithine, glycine and octadecanoic acid (Figure 4). To compare the differences in metabolic profiles of *A. sibiricum* infected with different *Epichloë* species, E samples were removed from the OPLS-DA model. Six differential metabolites were screened (VIP > 1, *p* < 0.05), and the contents of β-gentiobiose, ribonic acid and inositol in Es+ plants were significantly higher than those in Eg+ plants, while the opposite trends were observed for the contents of tyrosine, maltose and α-linolenic acid (Figure 5). The pathway analysis revealed that both Eg and Es infection mainly affected galactose metabolism, starch and sucrose metabolism, tyrosine metabolism (isoquinoline alkaloid metabolism was contained) and phenylalanine metabolism (Figure 6).

### 3.3. Correlation of Metabolites in Epichloë Endophytes and the Host A. sibiricum

In this study, except for mannitol produced by endophytes, 19 of the same metabolites existed in *Epichloë* endophyte and host *A. sibiricum*, accounting for 54.3% of the total number of metabolites in endophytes (Figure 7). We analyzed the correlation of metabolite contents between the variation in the two *Epichloë* species and the variation in *A. sibiricum* infected with these two *Epichloë* species. Es had significantly higher contents of tyrosine and α-linolenic acid than Eg, and the same significant differences in these two metabolites were also found between Es+ plants and Eg+ plants. There was no significant difference between Es and Eg in the contents of 13 metabolites, on which there was also no significant difference between the two *A. sibiricum*-*Epichloë* symbionts. In general, there was a significant positive correlation (*p* = 0.0420) between the metabolite contents of endophytes and the host (Figure 8).

## 4. Discussion

### 4.1. Effect of Epichloë Endophyte Infection on Metabolites of Host A. sibiricum

This study indicated that *Epichloë* infection had an obvious effect on the metabolism of the host *A. sibiricum*. A previous study on *L. perenne* indicated that *Epichloë* infection altered the expression pattern of more than 38% of genes in the host plant [49]. In this study, among the 39 metabolites identified in *A. sibiricum*, at least 11 metabolites were increased significantly by *Epichloë* infection. A main regulation was to increase the metabolites of several kinds of soluble sugars, causing the total soluble sugar contents in endophyte-infected plants to be approximately 66.3% higher than that in E− plants. Soluble sugars are not only direct precursors of energy supply substances and various metabolic reaction intermediates in plants but also signal molecules that control the expression of genes related to growth, metabolism and stress resistance [50,51]. In both galactose metabolism and starch-sucrose metabolism, which were potentially regulated by endophytes, sucrose is a key metabolite. The promoting effect on the accumulation of the main carbon sources and energy substances, such as sucrose, glucose, fructose and maltose, may give infected plants positive contribution in growth and morphogenesis. The soluble sugars increased by *Epichloë* endophytes can also be utilized as osmotic regulators to enhance the tolerance of host plants to drought stress and salt stress [11,34]. In addition, the content increase in glucose, maltose and allose may help to improve the resistance of host plants to oxidative damage caused by stresses [52,53,54,55,56].

Another main effect of *Epichloë* infection on metabolites of *A. sibiricum* was to increase aromatic amino acids (phenylalanine and tyrosine); a similar result was reported in a study on *L. perenne* [57]. Phenylalanine and tyrosine are biosynthetic precursors of phenolic compounds, which are the main antioxidants in plants, and phenolic compounds play an important role in resisting pathogens and herbivorous insects [58,59,60]. Previous studies have found that *Epichloë* infection can increase phenolic compound content in *L. perenne* and *A. sibiricum* [43,61]. Our study further confirmed that endophyte infection significantly increased α-tocopherol, whose biosynthetic precursor is tyrosine, and α-tocopherol can scavenge free radicals and protect biofilms from peroxide damage [62]. Moreover, *Epichloë* infection also increased α-linolenic acid in *A. sibiricum*, α-linolenic acid and its oxidation product oxylipins have anti-pathogenic activity [63,64], and α-linolenic acid can also participate in the plant defense response to pathogens and herbivorous insects [65]. Therefore, we speculate that the positive regulation of phenylalanine metabolism and tyrosine metabolism by *Epichloë* endophytes and the increase in α-linolenic acid may improve the antioxidant capacity of the host *A. sibiricum* and help the host resist biotic stresses.

### 4.2. Correlation of Metabolites between Epichloë Endophytes and the Host A. sibiricum

In this study, we found that the major direct effect of *Epichloë* infection was the presence of mannitol in *A. sibiricum*. Similar results were reported in studies on *L. perenne* and *F. arundinacea,* in which mannitol was produced by *Epichloë* endophytes [11,57]. Mannitol is an osmotic protective substance and scavenges reactive oxygen species (ROS) to reduce oxidative damage in plants [66,67], so it is likely to play a role in enhancing the resistance of the host *A. sibiricum* to stress.

We noticed that more than half of the metabolites in *Epichloë* endophytes also existed in the metabolic profiles of *A. sibiricum*; similarly, it has been found that 56% and 76% of the metabolites of two *Epichloë* strains, NEA12 and NEA23, were shared with the host *L. perenne* [68]. Among the 19 mutual metabolites in endophytes and *A. sibiricum*, Es had significantly higher contents of tyrosine and α-linolenic acid than Eg, and the same significant differences in these two metabolites were also found between Es+ plants and Eg+ plants. Furthermore, there was no significant difference between Es and Eg in the contents of 13 metabolites, on which there was also no significant difference between the two *A. sibiricum*-*Epichloë* symbionts. The positive correlation between endophyte metabolites and the host metabolites indicated that the effects of species difference of *Epichloë* endophytes on most of their own metabolites could also be endowed to the corresponding host grasses.

## 5. Conclusions

This study was based on the analysis of the difference in metabolites between *Epichloë*-infected and uninfected plants under normal growth conditions. *Epichloë* infection significantly affected the metabolism of the host *A. sibiricum*, and these changes may affect the growth and stress resistance of the host. It was reported that the beneficial effect of endophyte infection on the host was usually more obvious under stress [11,13]. In natural habitats, plants often face a variety of biological disturbances and abiotic stresses; as a result, endophyte infection may have a greater influence on the metabolism of host plants.

## Figures and Tables

**Figure 1 jof-08-00619-f001:**
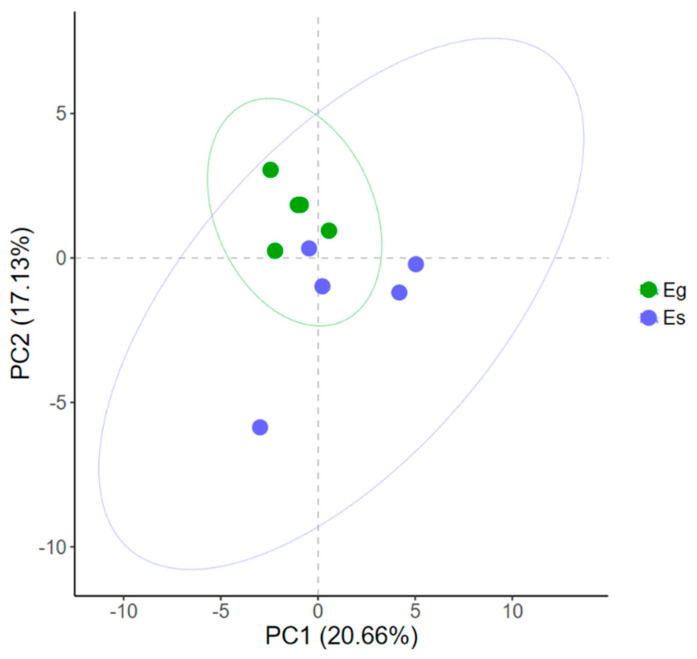
Principal component analysis (PCA) of the metabolic profiles of *Epichloë gansuensis* (Eg) and *Epichloë sibirica* (Es). The symbols refer to samples, and the ellipse defines the 95% confidence region.

**Figure 2 jof-08-00619-f002:**
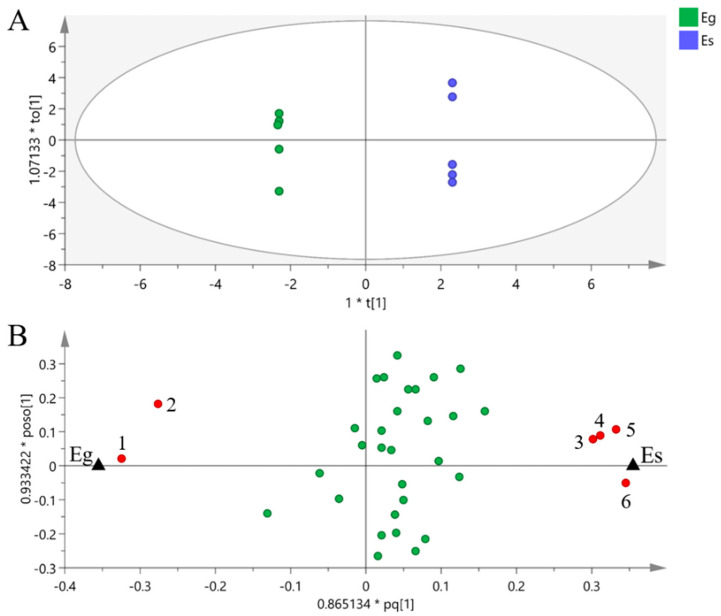
Orthogonal partial least squares discriminant analysis (OPLS-DA) of the metabolic profiles of *Epichloë gansuensis* (Eg) and *Epichloë sibirica* (Es). OPLS-DA: 1 + 5 + 0 latent variables, with R^2^X = 0.755, R^2^Y = 1.000, Q^2^ = 0.699. The ellipse in score plot (**A**) defines the 95% Hotelling’s T2 confidence region, and the symbols refer to samples. The symbols in loading plot (**B**) refer to compounds in metabolic profiles, the red symbols refer to differential metabolites screened between Eg and Es (VIP > 1, *p* < 0.05), (1) proline, (2) glutamic acid, (3) trans-9-octadecenoic acid, (4) tyrosine, (5) trehalose, (6) α-linolenic acid, and the green symbols refer to metabolites with no significant difference.

**Figure 3 jof-08-00619-f003:**
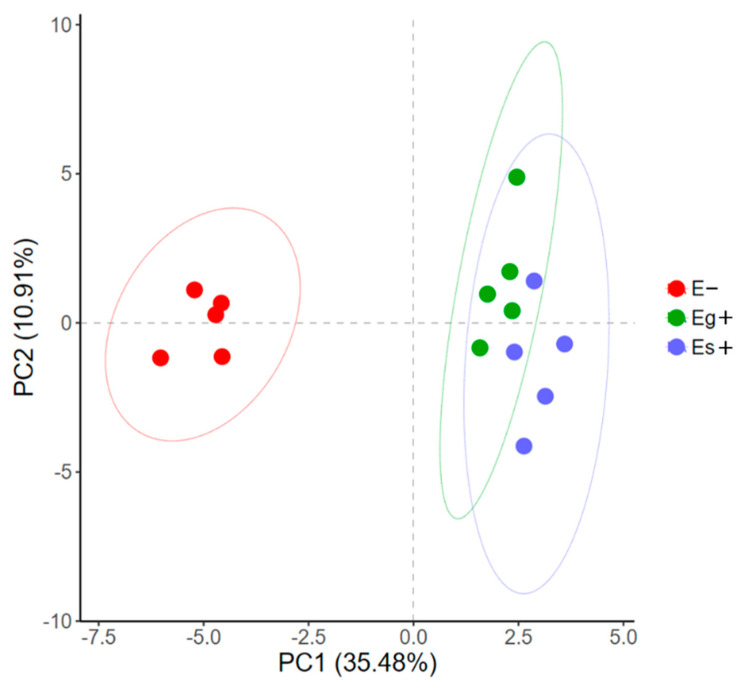
PCA of metabolites in *Achnatherum sibiricum* infected with *Epichloë gansuensis* (Eg+) and *Epichloë sibirica* (Es+) and without endophytes (E−). The symbols refer to samples, and the ellipse defines the 95% confidence region.

**Figure 4 jof-08-00619-f004:**
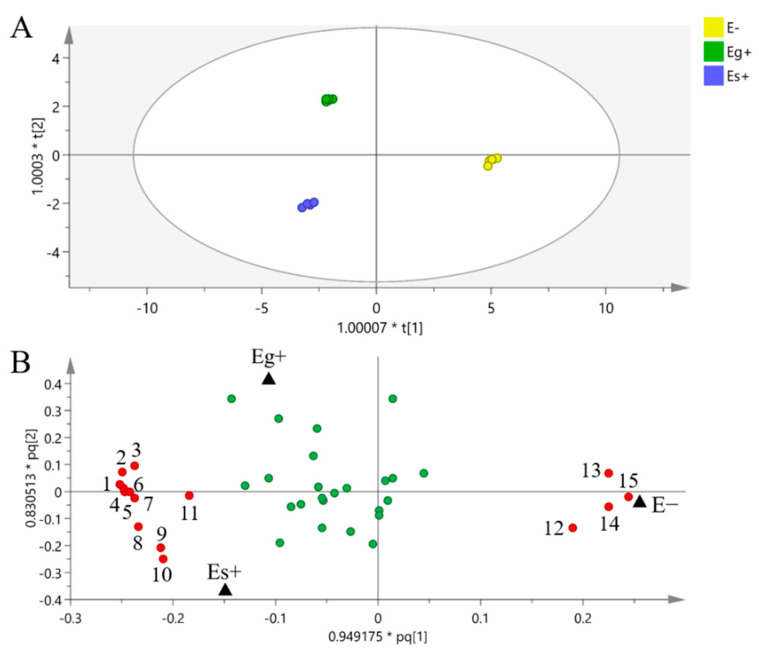
OPLS-DA of the metabolic profiles of *Achnatherum sibiricum* not infected with endophyte (E−) and infected with *E. gansuensis* (Eg+) and *E. sibirica* (Es+). OPLS-DA: 2 + 3 + 0 latent variables, with R^2^X = 0.648, R^2^Y = 0.998, Q^2^ = 0.843. The ellipse in score plot (**A**) defines the 95% Hotelling’s T2 confidence region. Symbols in the loading plot (**B**) refer to compounds in metabolic profiles, and the red symbols refer to differential metabolites screened between E−, Eg+ and Es+ plant (VIP > 1, *p* < 0.05), (1) mannitol, (2) sucrose, (3) threonic acid, (4) allose, (5) fructose, (6) glucose, (7) α-tocopherol, (8) tyrosine, (9) α-linolenic acid, (10) maltose, (11) phenylalanine, (12) glycine, (13) valine, (14) ornithine and (15) octadecanoic acid, and the green symbols refer to metabolites with no significant difference.

**Figure 5 jof-08-00619-f005:**
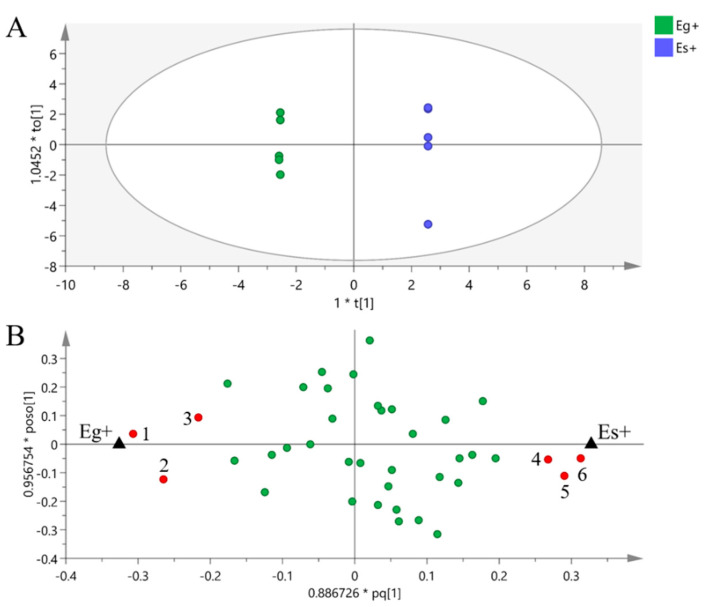
OPLS-DA of the metabolic profiles of *Achnatherum sibiricum* infected with *E. gansuensis* (Eg+) and *E. sibirica* (Es+). OPLS-DA: 1 + 4 + 0 latent variables, with R^2^X = 0.645, R^2^Y = 1.000, Q^2^ = 0.840. The ellipse in score plot (**A**) defines the 95% Hotelling’s T2 confidence region. Symbols in the loading plot (**B**) refer to compounds in metabolic profiles, and the red symbols refer to differential metabolites screened between Eg+ and Es+ plant (VIP > 1, *p* < 0.05), (1) ribonic acid, (2) inositol, (3) β-gentiobiose, (4) tyrosine, (5) α-linolenic acid and (6) maltose, and the green symbols refer to metabolites with no significant difference.

**Figure 6 jof-08-00619-f006:**
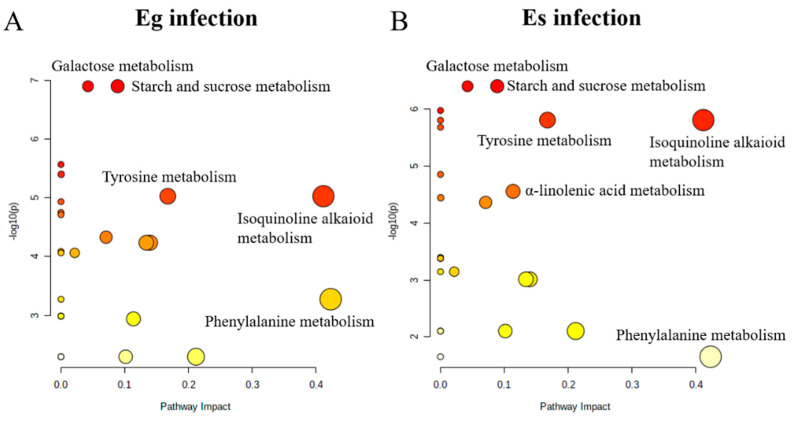
KEGG pathway analysis of the effect of (**A**) Eg infection and (**B**) Es infection on the metabolism of *A. sibiricum*. Each bubble represents a KEGG pathway, its size varies with the impact value, and its color varies with the enrichment significance. The pathway impact value represents the relative importance of metabolites in the pathway, reflecting the influence of the differential metabolites on this pathway. The vertical axis represents the enrichment significance of metabolites in the pathway, reflecting the credibility of the regulation in this pathway. The most related pathways regulated by endophytes are annotated.

**Figure 7 jof-08-00619-f007:**
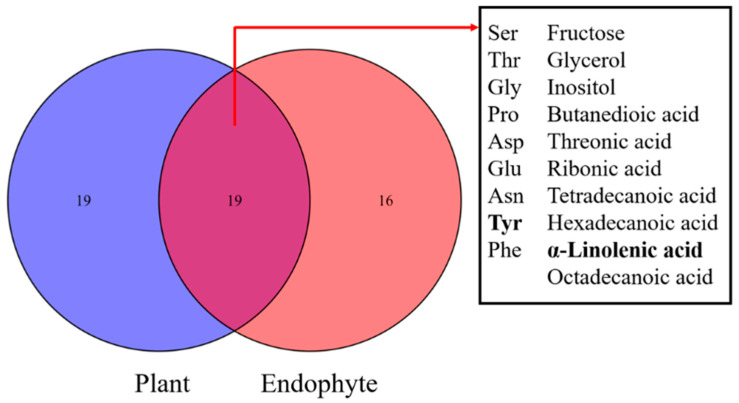
Venn diagram of the metabolite distribution in *Epichloë* endophytes and *A. sibiricum.* Metabolites present in both endophytes and *A. sibiricum* are listed, in which the metabolites with bold font are not only the differential metabolites of Eg and Es but also the differential metabolites of *A. sibiricum* infected with Eg and Es.

**Figure 8 jof-08-00619-f008:**
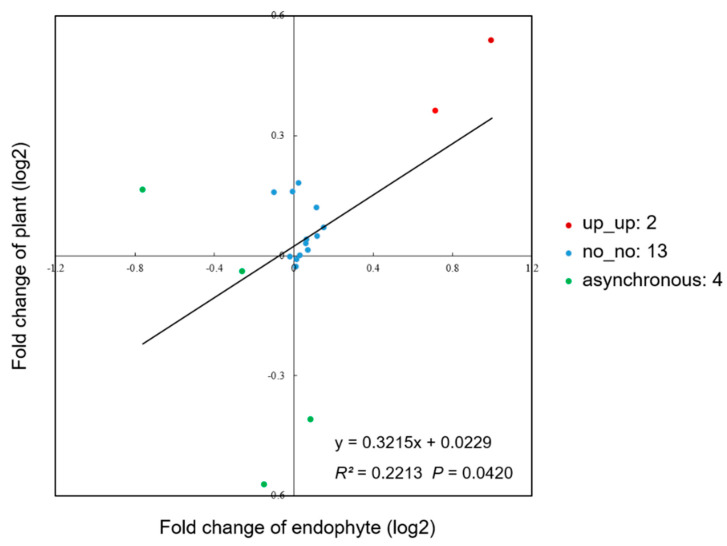
Scatter plot of correlation analysis of metabolites in *Epichloë* endophyte and *A. sibiricum*. The fold change of endophyte is the metabolite content ratio of Es to Eg, and the fold change of plant is the metabolite content ratio of Es+ plants to Eg+ plants. “up_up” refers to significant increase in both the endophyte and the plant, “no_no” refers to insignificant change in both endophyte and plant, “asynchronous” refers to asynchronous change between the plant and the endophyte.

## Data Availability

Not applicable.

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
