# Peer review of "Comparative Research on Metabolites of Different Species of Epichloë Endophytes and Their Host Achnatherum sibiricum"

_jof, 2022, doi:10.3390/jof8060619_

Round 1

Reviewer 1 Report

This manuscript describes the metabolites  found associated with infection of Achnatherum grass with two different Epichloe species. The English needs extensive editing. The primary problem with the paper is that it attempts to overextend the work to all grasses and fungal endophytes. It needs to be very specific in all mentions of this paper to include the single species of grass and the two species of Epichloe.

Line 16-17 - change to "Epichloe infection indirectly caused an increase in the soluble sugars in the Achnatherum host grass related to growth..."

Line 27 - delete "but do not cause obvious symptoms."

Line 33 - change to "...mutual benefit. Host plants..."

Line 43 - change "on" to "of"

Line 64 change to "While benefical fungal endophytes..."

Line 68 change to "...how beneficial fungal endophytes..."

Line 70 change to "...steppe, that commonly has 100% Epichloe natural infection rate [38]."

Line 71-73  change to "Two other species of Achnatherum are known to be infected with Epichloe; A. inebrians is distributed.... and A. robustum is distributed...., and both are associated with livestock poisoning [39,40]."

Line 74 change to "In contrast, A. sibiricum infected with Epichloe...."

Lines 75-86 - change "endophyte" to "Epichloe" and "host grass" to "Achnatherum"

Line 77 - delete " which show that the existence of the endophyte undoubtedly results in changes in multiple metabolites in A. sibiricum."

Lines 78-80 - change to "We chose to analyze the metabolites in Achnatherum sibiricum and Epichloe because this grass in Inner Mongolia naturally hosts either Epichloe ganseunsis (Eg) or Epchloe sibirica (Es) [44]."

Line 84 - change "expected to be answered" to "addressed"

Line 109 - change "repeats were set" to "replicates were tested"

Figure 3 - Please outline the yellow with black. The yellow PCA circle was particularly difficult to see.

Figure 6 - Please explain the significance of the colors in the legend. Need to define "enrichment significance" and "relative importance" in the legend also.

Discussion - Please change all "endophytes" to Epichloe" and "host grass" to "Achnatherum"

Line 245-246 - Present your conclusion first, then compare with the L. perenne/Epichloe data.

Line 307 - change "It was report that" to "We found"

Line 311 - Don't include this here. It is unsubstantiated since you did not present any evidence for this. If you want to include this, you will have to add a paragraph to the discussion with lots of supporting evidence.

Author Response

Response to Reviewer 1 Comments

Point 1: The English needs extensive editing. The primary problem with the paper is that it attempts to overextend the work to all grasses and fungal endophytes. It needs to be very specific in all mentions of this paper to include the single species of grass and the two species of Epichloë.

Response 1: We rechecked the English in the manuscript and corrected mistakes. We changed "endophyte" to "Epichloë " and "host grass" to "A. sibiricum" or "the host A. sibiricum" in sentences which need to be specific. (Please see the whole revised manuscript).

Point 2: Line 16-17 - change to " Epichloë infection indirectly caused an increase in the soluble sugars in the Achnatherum host grass related to growth..."

Response 2: We rephrased the description as suggested in the revised manuscript. (Please see line 16-17)

Point 3: Line 27 - delete "but do not cause obvious symptoms."

Response 3: We delete the description as suggested. (Please see line 25)

Point 3: Line 33 - change to "...mutual benefit. Host plants..."

Response 3: We changed the punctuation here as suggested. (Please see line 31)

Point 4: Line 43 - change "on" to "of"

Response 4: We corrected the word as suggested. (Please see line 40)

Point 5: Line 64 change to "While benefical fungal endophytes..."

Response 5: We rephrased the description as suggested. (Please see line 61)

Point 6: Line 68 change to "...how beneficial fungal endophytes..."

Response 6: We rephrased the description as suggested. (Please see line 65)

Point 7: Line 70 change to "...steppe, that commonly has 100% Epichloë natural infection rate [38]."

Response 7: We rephrased the description to "in the natural population, its Epichloë infection rate is commonly close to 100%". (Please see line 68-69)

Point 8: Line 71-73  change to "Two other species of Achnatherum are known to be infected with Epichloë; A. inebrians is distributed.... and A. robustum is distributed...., and both are associated with livestock poisoning [39,40]."

Response 8: We rephrased the description as suggested. (Please see line 68-70)

Point 9: Line 74 change to "In contrast, A. sibiricum infected with Epichloë...."

Response 9: We rephrased the description as suggested. (Please see line 71)

Point 10: Lines 75-86 - change "endophyte" to " Epichloë " and "host grass" to "Achnatherum"

Response 10: We changed "endophyte" to "Epichloë " as suggested, and changed "host grass" to "A. sibiricum" because "A. sibiricum" is more accurate to make the description be specific as suggested in Point 1. (Please see line 72-82)

Point 11: Line 77 - delete " which show that the existence of the endophyte undoubtedly results in changes in multiple metabolites in A. sibiricum."

Response 11: We deleted the sentence as suggested. (Please see line 74)

Point 12: Lines 78-80 - change to "We chose to analyze the metabolites in Achnatherum sibiricum and Epichloë because this grass in Inner Mongolia naturally hosts either Epichloë ganseunsis (Eg) or Epichloë sibirica (Es) [44]."

Response 12: We rephrased the description as suggested. (Please see line 74-75)

Point 13: Line 84 - change "expected to be answered" to "addressed"

Response 13: We rephrased the description as suggested. (Please see line 79)

Point 14: Line 109 - change "repeats were set" to "replicates were tested"

Response 14: We rephrased the description "5 biological repeats were set" to "each kind had 5 biological replicates for analysis", at reference of your suggestion. (Please see line 104-105)

Point 15: Figure 3 - Please outline the yellow with black. The yellow PCA circle was particularly difficult to see.

Response 15: We redrew the Figure 3 with the symbols and circle of E- samples changed to red to be clearly seen.

Point 16: Figure 6 - Please explain the significance of the colors in the legend. Need to define "enrichment significance" and "relative importance" in the legend also.

Response 16: The color of bubbles in Figure 6 indicates the enrichment significance of the metabolites in the pathway, it varies with the –log10(P) value. In the revised manuscript, we defined "enrichment significance" and "relative importance" in the legend.

Point 17: Discussion - Please change all "endophytes" to Epichloë" and "host grass" to "Achnatherum"

Response 17: We changed "endophyte" to "Epichloë " as suggested, and changed "host grass" to "A. sibiricum" in discussion. (Please see page 9-10)

Point 18: Line 245-246 - Present your conclusion first, then compare with the L. perenne/ Epichloë data.

Response 18: We added our conclusion before comparing with the L. perenne/ Epichloë data as suggested. (Please see line 246-247)

Point 19: Line 307 - change "It was report that" to "We found"

Response 19: The description "the beneficial effect of endophyte infection on the host was more obvious under stress" was concluded from previous studies, we added references at the end of the sentence. (Please see line 307-308)

Point 20: Line 311 - Don't include this here. It is unsubstantiated since you did not present any evidence for this. If you want to include this, you will have to add a paragraph to the discussion with lots of supporting evidence.

Response 20: We deleted this part in this paragraph as suggested. (Please see the last paragragh)

Reviewer 2 Report

The manuscript by Deng et al. reports a comparison of metabolites detectable by GC-MS between Epichloe infected and uninfected Achatherum sibiricum. The results are interesting, particularly the higher levels of several sugars in the endophyte infected plants. This will be of interest to other endophyte researchers. I have a few comments below.

  1. On line 108 they state that 5 biological replicates of the plant material were grown. Presumably that means 5 biological replicates were extracted and analyzed. This should be explicitly stated.
  2. Supplemental Table 1 lists the compounds detected in endophyte-infected plants and in the endophytes grown in culture but not the endophyte free plants. Would it be possible to add the endophyte free plants and show the analysis for the two endophyte types separately. Can the quantities and standard deviations detected for the 5 sample types be added and the compounds arranged so looking across the table the quantities could be compared.
  3. Lines 228 to 230 state that there was “in general a correlation between metabolite contents of endophytes and the host”. However, in the legend to Fig. 8 it looks like in general there was not a correlation – only two compounds were correlated. Does the correlation for the two compounds indicate that the endophytes were a major contributor to the amount seen in the E+ plants? How do the quantities in the endophyte-free plants compare? The authors should consider rewriting this section.
  4. The last paragraph in section 4.2 should be removed. The authors seem to be proposing that horizontal gene transfer may be the reason that the levels of the compounds tyrosine and alpha-linolenic acid were correlated in the plants and the endophytes in culture. These metabolites are those of basic cellular metabolism and are shared among most eukaryotes. Without more specific data it does not seem reasonable to invoke horizontal gene transfer to explain this observation.

Author Response

Response to Reviewer 2 Comments

Point 1: On line 108 they state that 5 biological replicates of the plant material were grown. Presumably that means 5 biological replicates were extracted and analyzed. This should be explicitly stated.

Response 1: We extracted and analyzed 5 biological replicates. To explicitly state this, we rephrased the description "5 biological repeats were set" to "each kind had 5 biological replicates for analysis". (Please see line 104-105)

Point 2: Supplemental Table 1 lists the compounds detected in endophyte-infected plants and in the endophytes grown in culture but not the endophyte free plants. Would it be possible to add the endophyte free plants and show the analysis for the two endophyte types separately. Can the quantities and standard deviations detected for the 5 sample types be added and the compounds arranged so looking across the table the quantities could be compared.

Response 2: We remade the Supplemental Table as you suggested. We added the data of endophyte free plants and showed the data of Eg-infected and Es-infected plants separately, and the data of endophytes and plants were showed in 2 tables separately, the quantities and standard of metabolites deviations were added into the revised Supplemental Tables.

Point 3: Lines 228 to 230 state that there was “in general a correlation between metabolite contents of endophytes and the host”. However, in the legend to Fig. 8 it looks like in general there was not a correlation – only two compounds were correlated. Does the correlation for the two compounds indicate that the endophytes were a major contributor to the amount seen in the E+ plants? How do the quantities in the endophyte-free plants compare? The authors should consider rewriting this section.

Response 3: In Figure 8, there was a significant positive correlation between endophyte metabolites and host metabolites in general due to the P value = 0.0420, We rewrote the Section 3.3 (Please see line 224-230), we redrew the Figure 8 and corrected the explaining in legend of Figure 8. Both Es+ plants and Eg+ plants had significantly higher contents of the 2 compounds (tyrosine and α-linolenic acid) than E- plants (Figure 4), indicating that the difference of endophyte species was a major contributor to the differences of metabolite contents between the two A. sibiricum-Epichloë symbionts.

Point 4: The last paragraph in section 4.2 should be removed. The authors seem to be proposing that horizontal gene transfer may be the reason that the levels of the compounds tyrosine and alpha-linolenic acid were correlated in the plants and the endophytes in culture. These metabolites are those of basic cellular metabolism and are shared among most eukaryotes. Without more specific data it does not seem reasonable to invoke horizontal gene transfer to explain this observation.

Response 4: We deleted those descriptions seem to propose that horizontal gene transfer may be the reason that endophyte metabolites and host metabolites were correlated. we quote this paper (Wang et al., 2020) to illustrate that endophyte infection might result in a closer correlation of metabolites with the host by changing the genome of host. (Please see line 295-302)

This manuscript is a resubmission of an earlier submission. The following is a list of the peer review reports and author responses from that submission.